# Serum Inflammatory and Oxidative Stress Markers in Patients with Vitiligo

**DOI:** 10.3390/jcm12185861

**Published:** 2023-09-09

**Authors:** Asma Kassab, Yassine Khalij, Yosra Ayed, Najla Dar-Odeh, Amal A. Kokandi, Meriam Denguezli, Monia Youssef

**Affiliations:** 1Biochemistry and Molecular Biology Laboratory, Faculty of Pharmacy, University of Monastir, Monastir 5019, Tunisia; khalij.yassine@gmail.com; 2Department of Fundamental Sciences, Faculty of Dental Medicine, University of Monastir, Monastir 5019, Tunisia; yosraayed82@gmail.com (Y.A.); myriam_denguezli2@yahoo.fr (M.D.); 3Department of Oral Surgery, Oral Medicine and Periodontics, School of Dentistry, The University of Jordan, Amman 11942, Jordan; najla_dar_odeh@yahoo.com; 4Department of Dermatology, Faculty of Medicine, King Abdulaziz University, Jeddah 21589, Saudi Arabia; 5Department of Dermatology, Hospital of Fattouma Bourguiba, Faculty of Medicine, University of Monastir, Monastir 5019, Tunisia; monia.youssef@gmail.com

**Keywords:** vitiligo, oxidative stress, antioxidant enzymes, IL-15, CXCL9, CXCL10, VASI, VIDA

## Abstract

Background: Vitiligo is a common chronic hypomelanotic skin disorder. An intricate pool of markers associated with a complex combination of biological and environmental factors is thought to be implicated in etiology. This study aims to investigate the most important markers associated with vitiligo pathogenesis, including redox status, inflammation, and immune profile, in patients with vitiligo. Materials and Methods: The study included a total of 96 subjects: 30 patients with active non-segmental vitiligo, 30 patients with stable non-segmental vitiligo, and 36 controls. The vitiligo area severity index (VASI) and vitiligo disease activity score (VIDA) were determined. The following serum parameters were assessed: antioxidant status (TAS), superoxide dismutase activity (SOD), catalase activity (CAT), glutathione peroxidase activity (GPx), glutathione-S-transferase activity (GST), malondialdehyde (MDA), advanced oxidation protein products (AOPP), C reactive protein (CRP), interleukin-15 (IL-15), and chemokines (CXCL9, CXCL10). Results: The VASI score was not significantly different between active and stable vitiligo patients, as it was approximately 0.1. TAS, CAT, GPx, and GST were significantly lower in vitiligo patients compared to controls (*p* < 0.05). They were also significantly lower in active vitiligo when compared to stable vitiligo (*p* < 0.05). However, SOD levels were significantly higher in vitiligo patients than in controls and in the active vitiligo group than in the stable vitiligo group (*p* < 0.05). MDA and AOPP levels were significantly higher in patients with active and stable vitiligo compared to controls (*p* < 0.05). However, they did not significantly differ between active and stable vitiligo patients (*p* < 0.05). In both active and stable vitiligo, CRP and IL-15 were significantly higher than controls (*p* < 0.05). Whereas CRP was significantly higher in active (range = 2.0–7.2, mean = 4.46 ± 1.09) than in stable vitiligo (range = 1.6–6.7, mean = 3.75 ± 1.08) (*p* < 0.05). There was no significant difference in IL-15 levels between active and stable vitiligo. In both active and stable vitiligo, CXCL9 and CXCL10 were significantly higher than controls (*p* < 0.05), and they were significantly higher in active than stable vitiligo (*p* < 0.05). Conclusions: In vitiligo, oxidative damage induces an increase in pro-inflammatory IL-15, which in turn promotes IFN-γ-inducible chemokines such as CXCL9 and CXCL10. Further, there seems to be a link between the VASI score and IL-15 levels. These data imply that inhibiting IL-15 could be a promising method for developing a potentially targeted treatment that suppresses the early interplay between oxidant stress and IL-15 keratinocyte production, as well as between resident and recirculating memory T cells.

## 1. Introduction

Vitiligo is a chronic hypomelanotic autoimmune skin disorder characterized by epidermal melanocyte impairment resulting in pigment dilution, white macules, and the formation of hypopigmented patches [1]. It is the most common depigmenting skin condition, affecting 0.5–2% of the adult and child population worldwide [2], with a slight female predilection [3,4]. A complex combination of biological and environmental factors is thought to produce an intricate pool of active molecules that induce vitiligo. The pathogenesis of the disease is thought to be related to the constant stress to which epidermal melanocytes are frequently subjected. Several external stressors, such as ultraviolet light, pollutants, and phenolic compounds, act as stimuli that target melanocytes and cause reactive oxygen species overproduction (ROS) [5]. While melanocytes utilize various processes that maintain redox homeostasis, antioxidants such as superoxide dismutase, catalase, and glutathione may oppose the effects of ROS [6]. Oxidative products in melanocytes and keratinocytes are the key points for initiating cytokine release, innate immunity activation, and immune cell migration. Chemotaxis mediated by ROS has a role in vitiligo development by allowing innate and adaptive immune cells, mainly melanocyte-specific T lymphocytes, to be active in lesional cells [7,8]. Chemotactic cytokines, CXCL9 (a monokine induced by interferon-gamma) and CXCL10 (protein 10 induced by interferon-gamma-INF-γ) are effective chemotactic factors for mobilizing immune cells to injury and inflammation sites through binding to their common receptor, CXCR3. They activate and recruit leukocytes such as T cells, eosinophils, monocytes, and natural killer cells to mediate immunological responses. These chemotactic cytokines are shown to play a very important role in mice with vitiligo. Whereas CXCL9 plays the role of a “recruit” signal, especially in the dermis, CXCL10 acts as a tethering signal that increases T cell activity and controls their movement [9]. This process enables immune cell homing and leukocyte trans-endothelial migration toward endothelium adhesion molecules of peripheral inflammatory cells [10]. However, the precise mechanisms underlying individual genetic predisposition, melanocyte auto-aggressiveness, and immunological tolerance failure pathways have not been clearly identified yet.

Individual antioxidant levels have been thoroughly investigated in vitiligo patients; however, scarce data on total antioxidant capacity is available [11]. Moreover, there is conflicting data on the estimated levels of antioxidants such as superoxide dismutase (SOD) and catalase in vitiligo patients. While some studies reported significantly lower levels of SOD in vitiligo patients than controls [12,13], a recent meta-analysis concluded that patients with active or stable vitiligo had higher levels of SOD than healthy controls [14]. Furthermore, specific therapeutic approaches that target pro-inflammatory cytokines have been explored to achieve re-pigmentation or stop depigmentation. However, these were associated with variable results and failed attempts on many occasions, which highlights the importance of further exploration of molecular mechanisms associated with vitiligo and evaluating new therapeutic opportunities [15] that help to improve prognosis and promote the quality of life of vitiligo patients [16].

Therefore, we conducted this study to evaluate oxidative stress markers as well as relevant inflammatory and immune markers in patients with stable and active vitiligo. The study also aimed to investigate a possible correlation between these markers and disease activity.

## 2. Material and Methods

### 2.1. Patients

This study included a total of 96 subjects, divided into 60 patients with non-segmental vitiligo and 36 healthy controls. Among the patient group, 30 patients had active disease and 30 patients were stable. Patients attending the outpatient dermatology department of Fattouma Bourguiba Hospital at Monastir, Tunisia, were recruited on a consecutive basis during the study period from 2020 to 2022. Exclusion criteria for the patient group were vitiligo patients under 20 years old, history of smoking or alcohol drinking, patients on medications, patients who had other medical histories or coexisting diseases, and patients with a sign of infection. All included patients did not receive any medications or other therapeutic options, such as phototherapy, for at least one month prior to the start of the study. The control group consisted of age- and sex-matched healthy controls who were recruited from healthcare staff members. Similar exclusion criteria were employed, including smoking, alcohol drinking, systemic diseases, and infections.

The study protocol was approved by the institute’s Ethics Commission (approval no. 44/2018). All participants signed an informed written consent in adherence to the Declaration of Helsinki principles.

### 2.2. Clinical Assessment

The vitiligo area severity index (VASI) was used for clinical assessment. This index is based upon measuring the vitiligo skin surface in terms of hand units, whereby one hand unit corresponds to 1% of the total skin area. Table 1 presents the interpretation of the VASI scale used for clinical assessment.

The vitiligo disease activity score (VIDA) was used for clinical assessment of disease activity. Table 2 presents the interpretation of the VIDA score.

### 2.3. Blood Sampling

Peripheral blood samples of study participants were drawn and placed in two types of tubes: one containing lithium heparin and the other containing ethylene-diamine-tetra-acetic acid (EDTA). The plasma placed in lithium heparin tubes was used to measure antioxidant parameters (TAS, SOD, CAT, GPx, GST) and CRP, while EDTA-tube blood was used to measure oxidant stress products (MDA, AOPP), cytokines, and chemokines (IL-15, CXCL9, CXCL10). The samples were separated into aliquots and kept at −80 °C until analysis.

### 2.4. Assessment of Oxidative Stress Markers

TAS was assessed according to the method in Miller et al. [17] using the RANDOX manual kit as indicated by the manufacturer (Randox Laboratories Ltd., Crumlin, County Antrim, UK). In the TAS assay, 2,2′-Azino-di-3-ethylbenzthiazoline sulphonate is incubated with peroxidase and H_2_O_2_ to produce a free cation. The latter has a stable green color measured at 600 nm. When serum is added to the test tube, antioxidants inhibit the oxidative reaction, causing a suppression of the green color proportional to their concentration.

According to Arthur and Boyne [18], SOD activity was determined using the diagnostic kit RANSOD produced by RANDOX (Randox Laboratories Ltd., Crumlin, County Antrim, UK). The serum was treated with xanthine and xanthine oxidase to produce superoxide radicals that created a red formazan dye. The degree of inhibition of this process was subsequently used to determine superoxide dismutase activity. Under the circumstances of the experiment, one unit of SOD induced a 50% inhibition of the rate of reduction of radicals. SOD activity was expressed in U of SOD/10 mg of protein.

According to Paglia and Valentine [19], GPx activity was evaluated using the diagnostic kit RANSEL manufactured by RANDOX (Randox Laboratories Ltd., Crumlin, County Antrim, UK) and expressed in U of GPx/mg of protein. The protein concentration was determined using the Biuret technique.

CAT in erythrocytes was measured using a spectrophotometric method described by Aebi [20]. The CAT activity was determined at 25 °C by measuring H_2_O_2_ dissociation at 240 nm. The enzyme reaction rate was calculated as the quantity of H_2_O_2_ converted into H_2_O and ½ O_2_ for one minute. GST was determined using a spectrophotometer method as described by Habig et al. [21]. Potassium phosphate buffer (pH 7.5, containing 1 mM GSH and 1% absolute ethanol) and 1 mM 1-chloro-2,4-dinitrobenzene (CDNB) were added to plasma samples. The variation of the absorbance was measured at 340 nm. The results were expressed as the number of μM of GSH-CDNB conjugate produced per minute per milliliter.

Serum MDA was determined using a spectrophotometric method described by Yoshioka et al. [22]. The serum sample was incubated in a boiling water bath with 0.67% thiobarbituric acid (Sigma Aldrich, Burlington, MA, USA). 1,1,3,3′-tetramethoxy propane (Sigma Aldrich) was used as a standard. Absorbance was read at 535 nm by means of a Hitachi U-2900 spectrophotometer. MDA concentrations were deduced from a standard curve.

AOPP was determined using a method described by Witko-Sarsat et al. [23]. A solution of 1.16 M potassium iodide (Sigma Aldrich) and 10% acetic acid (Merck Millipore, Darmstadt, Alemanha) was added to plasma samples. The immediate measure was performed at 340 nm against 0.9% NaCl as a blank. Chloramine T (Sigma Aldrich) was used as the standard. AOPP concentrations were deduced from a standard curve.

### 2.5. Assessment of Inflammatory and Autoimmune Markers

CRP was measured using the immunoturbidimetric method. CRP-antibody interactions result in the assembly of immune complexes. The produced turbidity was spectrophotometrically quantified using an auto-analyzer (Konelab 20i, ThermoScientific, Waltham, MA, USA).

IL-15 was measured by means of an enzyme-linked immunosorbent assay (ELISA) according to the manufacturer’s recommendation (Catalogue number E0097Hu; Bioassay Technology Laboratory, Shanghai, China). Samples and standards were manipulated using the product’s guidelines and the quantitative sandwich enzyme immunoassay method. An ELISA reader (Stat fax) was used to read the color variation at 450 nm. Sera IL-15 in pg/mL levels were deduced from the standard curve.

Human chemokine kits were used to assess serum CXCL9 and CXCL10 (Catalogue number 552990; PeproTech, Neuilly-sur-Seine, France). The BD CBA human chemokine kits were used according to the instruction manual to quantify the levels of monokine-induced by Interferon-γ (CXCL9/MIG) and Interferon-γ-induced Protein-10 (CXCL10/IP-10) in EDTA-treated serum. The standards were reconstituted and serially diluted before mixing with the capture beads and the PE detection reagent. The assay diluent was used as the negative control for 0 pg/mL. The capture beads were bottled individually. The bead reagents (A1–A5) were immediately pooled before mixing them together with the PE detection reagent, standards, and samples. Monoclonal antibody microbeads against chemokines were quantified using a FACSVerse flow cytometer (Beckman Coulter, Inc., Brea, CA, USA). Each capture bead has an average of 300 recorded events. The analysis has been conducted by BD FCAP Array 3.0 software (Becton Dickinson and Company, San Jose, CA, USA). The values were expressed in picograms per milliliter.

### 2.6. Statistical Analysis

The statistical analyses were conducted using SPSS 23.0 (SPSS Science, Chicago, IL, USA). Numbers and percentages were used to represent categorical variables. The Kolmogorov–Smirnov test was used to establish the distribution’s normality. Range, median, mean, and standard deviation were used to express quantitative values. For normally distributed quantitative data, comparisons between study groups were performed using the Student’s *t*-test and the analysis of variance (ANOVA) test. For qualitative values, the chi-square test was used. The Pearson correlation coefficient was used to evaluate the correlation between the assessed parameters. *p* values of less than 0.05 were regarded as significant.

## 3. Results

A total of 96 subjects were included in the study. Their mean age was approximately 34 years, and females constituted 55% of the patient and control groups. The demographic (age and gender) and clinical characteristics (BMI and disease attributes) of the study sample are displayed in Table 3.

All the cases were non-segmental vitiligo, including acrofacial and generalized vitiligo. The BMI mean was comparable between the patient and control groups. The mean value of VASI was 9.40%. According to the VIDA scale, 50% of the patients were in the active stage and 50% were in the stable stage.

### Oxidative Stress Markers

Table 4 displays the antioxidant parameters of active vitiligo, stable vitiligo, and controls.

The VASI score was not significantly different between active and stable vitiligo patients as it was approximately 0.1.

TAS, CAT, GPx, and GST were significantly lower in vitiligo patients compared to controls (*p* < 0.05). They were also significantly lower in active vitiligo when compared to stable vitiligo (*p* < 0.05). However, SOD levels were significantly higher in vitiligo patients than in the controls and in the active vitiligo group than in the stable vitiligo group (*p* < 0.05).

MDA and AOPP levels were significantly higher in active and stable vitiligo compared to controls (*p* < 0.05). However, they did not significantly differ between active and stable vitiligo patients (*p* < 0.05) (Table 4).

In both active and stable vitiligo, CRP and IL-15 were significantly higher than controls (*p* < 0.05). Whereas CRP was significantly higher in active (range = 2.0–7.2, mean = 4.46 ± 1.09) than in stable vitiligo (range = 1.6–6.7, mean = 3.75 ± 1.08) (*p* < 0.05). There was no significant difference in IL-15 levels between active and stable vitiligo patients (Figure 1).

IL-15 was significantly correlated with VASI (r = 0.62, *p* < 0.05, Figure 2).

Figure 3 presents chemokine concentrations in the study sample. In both active and stable vitiligo, CXCL9 and CXCL10 were significantly higher than controls (*p* < 0.05). They were significantly higher in active than stable vitiligo (*p* < 0.05).

## 4. Discussion

The chronic course of vitiligo with alternate active and stable periods is challenging for clinicians, making it difficult to provide suitable and sustainable treatment options. Recent studies suggest that specific biomarkers can be estimated to evaluate vitiligo disease activity and response to treatment, thereby enhancing the prognosis and increasing the possibility of success. This study investigated specific markers related to vitiligo pathogenesis as redox status (TAS, SOD, GPx, CAT, MDA, and AOPP), inflammatory profile (CRP), innate immunity (IL-15), and adaptive immunity (CXCL9 and CXCL10) markers in patients with non-segmental vitiligo, which is the most prevalent type of vitiligo in comparison to healthy controls. Strict inclusion criteria were implemented to investigate the possible correlation between vitiligo and these markers. The control group, which was matched by age and sex, also had comparable BMI mean values. The previous literature confirmed that there is no association between BMI and vitiligo [24,25]. However, a recent systematic review suggested an association between vitiligo and the metabolic syndrome [26].

A reduced antioxidant capacity was identified in vitiligo patients in this study in the form of decreased TAS, CAT, GPx, and GST. This is consistent with other studies that also detected a decrease in TAS, CAT, GPx, and GST in vitiligo patients [11,27,28,29]. The decreased levels of these parameters imply that either an extended ROS is generated by exogenous or endogenous stimuli or that vitiligo patients have TAS levels that are insufficient for neutralizing the melanocyte-damaging oxidants [11]. A reduced GST is likely to be associated with mRNA suppression, as shown by the reduced GST expression in cultured keratinocytes from normal and lesional areas [29]. The reduced CAT levels and associated lower antioxidant activity will eventually produce more free radicals, necessitating increased levels of SOD to scavenge these radicals [11,30]. Furthermore, because melanin has an antioxidant function, it may influence SOD activity [15]. Likewise, when melanocyte apoptosis was induced, ROS increased and inhibited melanogenesis. In vitro tests showed that upon exposure of melanocytes to H_2_O_2_, the melanogenesis process was impeded [31]. Several toxic compounds are thought to be produced during melanogenesis in vitiligo, resulting in H_2_O_2_ buildup that inhibits catalase function and promotes melanocyte eradication.

The insufficient antioxidant protection in vitiligo patients also explains the increased oxidative damage in the form of a rise in oxidative damage markers, MDA and APPO [13]. AOPP is produced by the interaction of free radicals, specifically hydroxyl radicals, with amino acids. The mean value of AOPP was significantly higher in patients with vitiligo when compared to the controls. When AOPP accumulates inside cells, it increases the likelihood of organelle dysfunction. On the other hand, MDA is the end product of polyunsaturated fatty acid combustion generated by ROS. MDA levels were significantly elevated in vitiligo patients, indicating that the peroxidation process has been triggered and that this could affect cell membrane breakdown. The re-stabilization of the redox balance in patients with vitiligo by smearing narrowband UVB decreases MDA and increases GPx activity [32]. The emergence of these indicators, MDA and AOPP, reveals the presence of advanced oxidant stress, which can trigger transformations in the endoplasmic reticulum of melanocytes, culminating in the accumulation of defective proteins and the genesis of the “unfolded protein response” [33].

On the contrary to CRP, which was significantly higher in active than stable vitiligo patients, IL-15 showed no significant difference. Current results are consistent with some of the earlier findings [34]. The installed oxidative stress in the patients might be associated with the impairment of the keratinocyte inflammatory response, which would activate acquired immunity by increasing pro-inflammatory interleukin IL-15 release. Redox imbalance is believed to be the leading cytotoxic mechanism, resulting in varied kerato-melanocyte cytokine production. The initial action of an altered Th1 pattern associated with imbalanced type-1 cytokine IFN-γ expression is emerging in vitiligo pathogenesis.

In vitiligo pathogenesis, the role of IL-15 is represented by modulating IL-17 release and sustaining T cell memory actions [35]. T cell memory cells penetrate the epidermis and convey vitiligo maintenance by generating IFN γ and TNF α, therefore resulting in a cytotoxic action against melanocytes [35]. Previous research has also concluded that keratinocyte expression of IL-15 was substantially elevated in vitiligo patients and was highly related to H_2_O_2_ levels [36].

In our study, no significant difference in IL-15 was detected between active and stable vitiligo; however, it was significantly higher in vitiligo patients than in healthy controls. Further, IL-15 was significantly correlated with the VASI score. This was the most significant correlation established in the current study. This finding may indicate that IL-15 contributes to disease progression. IL-15 plays a key role in the formation of memory T cells; therefore, it was suggested that targeting IL-15 signaling may reverse disease [37] by using therapies such as multiple monoclonal antibodies [38].

CXCL9 and CXCL10 levels were significantly higher in patients (active and stable groups) than controls, and among the vitiligo patients, CXCL9 and CXCL10 levels were significantly higher in active than stable groups. The oxidative damage produced in epidermic cells boosts the production of IFN α, which in turn increases the expression of CXCL10, enhancing, therefore, lymphocyte homing [39]. Studies have reported that IFN α enhances CXCL9 and CXCL10 expression by keratinocytes [40]. These markers are linked to the diffusion of melanocyte-specific cytotoxic T lymphocytes into the epidermis basal layer to target melanocytes, causing melanin inadequacy [41]. IFN γ promotes skin homing of cytotoxic melanocyte CD8+ T lymphocytes through the induction of CXCL9 and CXCL10 by stressed keratinocytes [42], resulting in T cell propagation and CXCR3 expression on their membranes [43]. On the other hand, mature CD8+ T cells destroy melanocytes, executing the autoimmune reaction responsible for vitiligo [44].

Rashighi et al. have reported an elevated level of CXCL10 in both the serum and the lesional skin of mice with vitiligo [45]. Therefore, it is thought that CXCL10 upregulation enhances autoreactive T cell localization in the epidermis and is required for skin depigmentation development and maintenance in a vitiligo mouse model [45]. Furthermore, in animal models, it was shown that CXCL10 neutralization inhibits the emergence of new lesions and activates the re-pigmentation of previously depigmented regions [46].

Despite the comprehensive nature of this study in addressing a wide range of markers that are implicated in vitiligo initiation and progression, it has limitations. More accurate conclusions could be drawn if samples were directly obtained from non-lesional, peri-lesional, and lesional cells rather than utilizing serum samples only. Most of our active vitiligo patients had a VIDA score around +4, so it would be more informative to include a larger sample with longer disease courses in future studies.

On the other hand, this study is among the first to report new findings about the pathogenesis of vitiligo and the role of cytokines in its etiology. This should pave the way for tailoring novel therapeutic approaches for treating this chronic, intractable disease.

## 5. Conclusions

The redox state imbalance is more prominent in active vitiligo than stable vitiligo, as demonstrated by lower TAS, antioxidant enzyme activities (CAT and GPx), and higher oxidative products (MDA and AOPP). The activity of SOD increases to compensate for an altered cellular redox state. Furthermore, higher levels of CRP, IL-15, CXCL9, and CXCL10 are identified in active vitiligo. Collectively, oxidative stress imbalances promote IFN-inducible chemokines by upregulating pro-inflammatory IL-15. Interestingly, the VASI score and IL-15 levels are correlated. Based on these findings, blocking IL-15 could be a promising approach to developing targeted treatments that inhibit the early interaction between oxidative stress and the release of IL-15 by keratinocytes and between recirculating memory T cells and T cell memory cells.

Figure 4 presents a graphical conclusion for the pathogenesis of vitiligo according to this study.

## Figures and Tables

**Figure 1 jcm-12-05861-f001:**
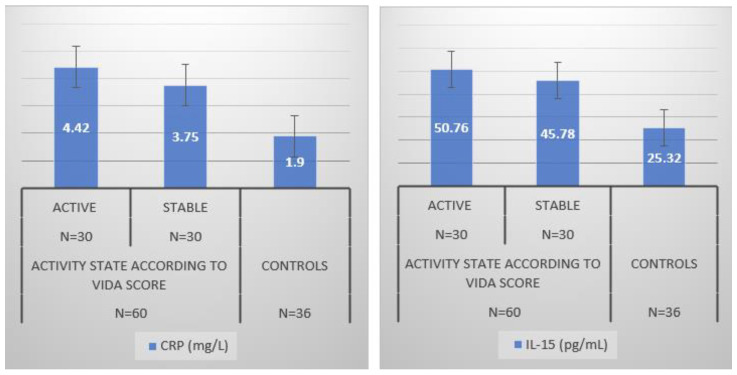
Inflammatory parameters of the population study. CRP: C-Reactive Protein; IL-15: Interleukin 15.

**Figure 2 jcm-12-05861-f002:**
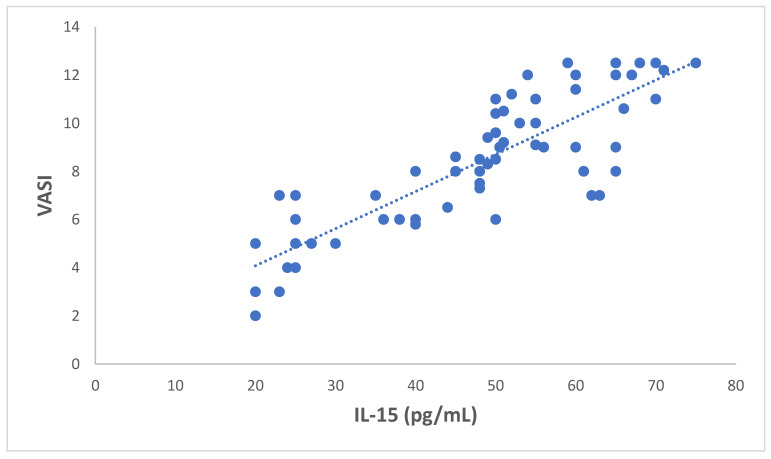
Correlation between serum levels of IL-15 (pg/mL) and VASI in patients with vitiligo (*n* = 60). r = 0.62, *p* < 0.05.

**Figure 3 jcm-12-05861-f003:**
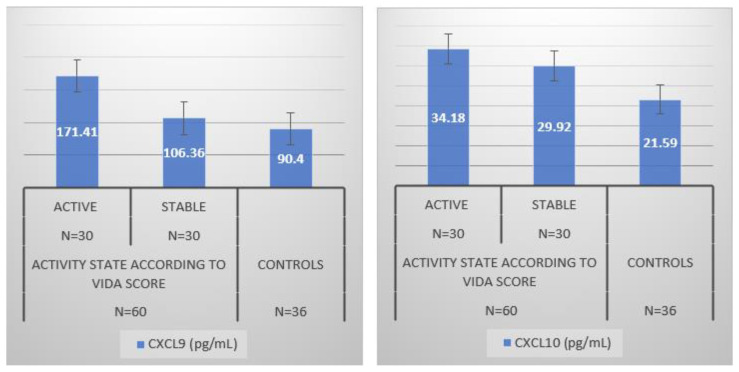
IFN-gamma-inducible chemokines in the study population. CXCL9: monokine induced by interferon-gamma; CXCL10: protein 10 induced by interferon-gamma.

**Figure 4 jcm-12-05861-f004:**
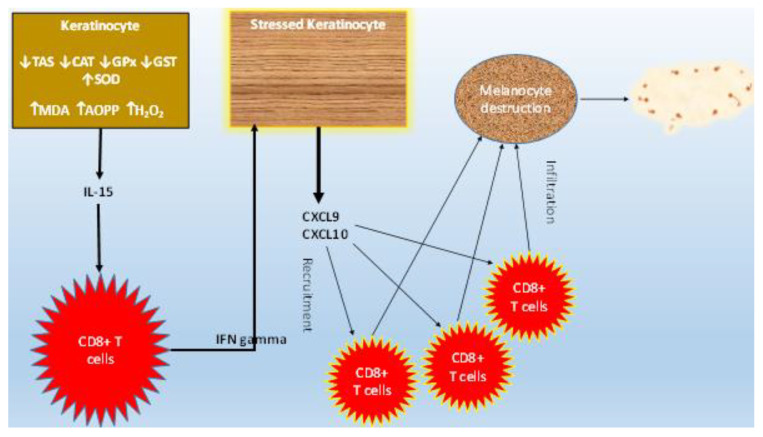
Graphical conclusion. Increased oxidative damage mediates the upregulation of pro-inflammatory IL-15, enhancing IFN-γ-inducible chemokines CXCL9 and CXCL10 secretion.

**Table 1 jcm-12-05861-t001:** VASI scale used for clinical assessment of disease severity.

Units	Clinical Interpretation
100%	Complete depigmentation
90%	Pigment specks
75%	Depigmented areas exceed pigmented areas
50%	Pigmented and depigmented areas are equal
25%	Pigmented areas exceed depigmented areas
10%	Only depigmentation specks are present

VASI = vitiligo areas (hand units) × depigmentation extent

**Table 2 jcm-12-05861-t002:** VIDA scale presents the six grades used to assess the vitiligo stage.

Grade	Disease Activity	Stage
+4	≤6 weeks	Active
+3	6 weeks–3 months	Active
+2	3–6 months	Active
+1	6–12 months	Active
0	Stable for ≥one year	Stable
−1	Stable with spontaneous repigmentation	Stable

**Table 3 jcm-12-05861-t003:** Demographic and clinical data of the study population.

	Vitiligo Patients *n* = 60	Controls *n* = 36
Age (mean ± SD, years)	34.5 ± 10	34.1 ± 13
Gender (female/male)	33/27	20/16
BMI (mean ± SD, Kg/m^2^)	25.7 ± 4.5	26.0 ± 3.5
Disease duration (Mean ± SD, years)	5.6 ± 4.3	-
Median [min–max]	6 [1–12]	
VASI scores (%)		
Mean ± SD	9.40 ± 2.4	
Median [min–max]	5.5 [2–12.5]	
VIDA score		
Mean ± SD	1.71 ± 1.2	
Median [min–max]	3 [−1–4]	
VIDA scale, *n* (%)		
+4	24 (40%)	
+3	2 (3.33%)	
+2	2 (3.33%)	
+1	2 (3.33%)	
0	25 (41.66%)	
−1	5 (8.33%)	

SD: Standard Deviation; BMI: Body mass index; VASI: vitiligo area severity index; VIDA: vitiligo disease activity score.

**Table 4 jcm-12-05861-t004:** Antioxidant status in controls, active, and stable vitiligo patients.

	Clinical Stage		Controls *n* = 36	P_1_	P_2_	P_3_
	Active (*n* = 30)	Stable (*n* = 30)				
VASI Score						NS
Mean ± SD	9.50 ± 2.5	8.80 ± 3.40				
Median [Range]	7.5 [3–12.2]	7 [2–12.5]				
**Antioxidant parameters**						
TAS (mmol/L)				0.005	0.040	0.050
Mean ± SD	1.70 ± 0.45	1.97 ± 0.19	2.05 ± 0.22			
Median [range]	1.56 [1.15–2.25]	1.80 [1.70–2.35]	1.90 [1.80–2.35]			
SOD (U/mg)				0.001	0.020	0.030
Mean ± SD	3.70 ± 0.75	3.57 ± 1.12	3.15 ± 1.02			
Median [range]	4 [2.50–4.5]	4.20 [2.40–4.70]	3 [2.10–4.20]			
CAT (U/mg)				0.001	0.020	0.040
Mean ± SD	34.75 ± 3.5	36.33 ± 4.02	38.5 ± 3.1			
Median [range]	35 [31–38.40]	37 [32.25–40.4]	38 [35–41.60]			
GPx (U/mg)				0.001	0.020	0.040
Mean ± SD	1.72 ± 0.85	1.89 ± 0.90	2.20 ± 0.44			
Median [range]	1.75 [0.85–2.70]	1.80 [0.95–2.80]	2.08 [1.76–2.70]			
GST (UI/mL)				0.001	0.001	0.020
Mean ± SD	18.12 ± 4.1	18.94 ± 3.5	20.00 ± 1.23			
Median [range]	18 [14–22]	19 [15–23]	19.5 [18–22]			
**Oxidant damage parameters**						
MDA (mmol/mL)				0.000	0.000	NS
Mean ± SD	5.30 ± 1.00	4.90 ± 2.10	4.50 ± 1.20			
Median [range]	4.50 [3.90–8.50]	4.50 [3.10–8.50]	4.10 [3.00–7.00]			
AOPP (umol/L)						
Mean ± SD	250.15 ± 200.20	205.36 ± 200.10	100.45 ± 50.18	0.000	0.000	NS
Median [range]	215.30 [70.80–845.30]	217.90 [65.65–850.40]	95.55 [70.20–250.30]			

TAS: Total Antioxidant Status, GST: Glutathione S-Transferase, SOD: Superoxide dismutase, CAT: Catalase, GPx: Glutathione peroxidase, MDA: Malondialdehyde, AOPP: Advanced oxidation protein products. Results are provided as mean ± standard deviation, P_1_: Active vitiligo vs. Controls, P_2_: Stable vitiligo vs. Controls, P_3_: Active vitiligo vs. stable vitiligo.

## Data Availability

Data supporting reported results can be obtained from corresponding author upon appropriate request.

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
