# Peer review of "Serum Inflammatory and Oxidative Stress Markers in Patients with Vitiligo"

_jcm, 2023, doi:10.3390/jcm12185861_

Round 1

Reviewer 1 Report

In the current research article, authors have explored "Serum Inflammatory And Oxidative Stress Markers In Patients With Vitiligo: A Case-Control Study". The subject is of interest and falls in the topics of “Journal of Clinical Medicine” Journal. 

After reviewing the manuscript thoroughly, I have following comments:

Why you select case control study?

Why only serum inflammatory and oxidative stress markers chosen for analysis?

Only 96 patients have been chosen. Explain.

The plasma placed in lithium heparin tubes was used to measure antioxidant parameters (TAS, SOD, CAT, GPx, GST) and CRP, while EDTA tube blood was used to measure oxidant stress products (MDA, AOPP), cytokines, and chemokines (IL-15, CXCL9, CXCL10). What is reason behind to select the lithium heparin tubes and EDTA tubes? Explain.

In statistical analysis, write the software along with version, used for analysis.

Abstract should be improved.

Novelty of the study must be added in the introduction part of manuscript.

Check the abbreviations throughout the manuscript and insert them where required.

There are number of grammatical and spelling mistakes in the manuscript. Check throughout the manuscript.

The English language needs to improve throughout the manuscript.

There are a number of grammatical and spelling mistakes in the manuscript. Check throughout the manuscript.

The English language needs to improve throughout the manuscript.

Reviewer 2 Report

Summary:

The authors performed a case-control study regarding serum inflammatory and oxidative stress markers in patients with vitiligo. The manuscript is interesting and well written. However, there are still some points need to be clarified.

Major concern:

1.          Did you set any washout period criteria for patients who have previously received medication?

2.          According to previous study, narrow band ultraviolet B may have an impact on the oxidative stress in patients with vitiligo (Said ER, Nagui NAER, Rashed LA, Mostafa WZ. Oxidative stress and the cholinergic system in non-segmental vitiligo: Effect of narrow band ultraviolet b. Photodermatol Photoimmunol Photomed. 2021 Jul;37(4):306-312). Did any case in the study receive phototherapy for treatment of vitiligo?

3.          There is selection bias in the matching of healthy control because the characteristics of healthcare staff members may be different from other patients.

Minor concern:

1.          Table 3, suggest adding the data of active and stable non-segmental vitiligo cases separately.

2.          Did VASI score corelated with other factors, such as antioxidant parameters, oxidant damage parameters, or other chemokines?

3.          It is recommended that using error bars to represent the standard deviation in Figures 1 and 3.

Some grammatical errors should be corrected.

Reviewer 3 Report

This presents new findings and is of interest for all experts in this field. I congratulate the authors to their work

Could you maybe provide the range of crp results? Although there is a significant difference between active and not active vitiligo, it seems to me as a very small difference. it could maybe be falsified if one patient had a very high level due to other reason for inflammation 

Reviewer 4 Report

The control group is uneven and too small for a case-control study. The methodology of the study has gross flaws. No BMI is known from the control group and study group. BMI plays a significant role in autoimmune diseases.

Extensive editing of English language required

Round 2

Reviewer 1 Report

In the present manuscript, authors have explored "Serum Inflammatory And Oxidative Stress Markers In Patients 2 With Vitiligo". The subject is of interest and falls in the topics of “Journal of Clinical Medicine” Journal. English can be improved. 

In the present manuscript, authors have explored "Serum Inflammatory And Oxidative Stress Markers In Patients 2 With Vitiligo". The subject is of interest and falls in the topics of “Journal of Clinical Medicine” Journal. All things are well explored in this paper. English can be improved. The study is acceptable in its present form.

Author Response

Reviewer 1 Round 2

In the present manuscript, authors have explored "Serum Inflammatory And Oxidative Stress Markers In Patients 2 With Vitiligo". The subject is of interest and falls in the topics of “Journal of Clinical Medicine” Journal. All things are well explored in this paper. English can be improved. The study is acceptable in its present form.

Response: Thank you for your encouraging remarks. The paper was revised for English language according to your recommendation. Edited areas are highlighted in blue.

Reviewer 2 Report

Minor concern:

It is recommended replacing orange column with error bars to represent the standard deviation in Figures 1 and 3.

Author Response

Reviewer 2 Round 2

It is recommended replacing orange column with error bars to represent the standard deviation in Figures 1 and 3.

Response: Figures 1 and 3 are now modified so that the orange columns are replaced by error bars representing the standard deviation

Reviewer 4 Report

All comments have been answered and changed by the authors. I have no further comments.

The language is easy to understand and comprehensible.

Author Response

Reviewer 4 Round 2

All comments have been answered and changed by the authors. I have no further comments.

Response: Thank you.